# Cultivation and Imaging of *S. latissima* Embryo Monolayered Cell Sheets Inside Microfluidic Devices

**DOI:** 10.3390/bioengineering9110718

**Published:** 2022-11-21

**Authors:** Thomas Clerc, Samuel Boscq, Rafaele Attia, Gabriele S. Kaminski Schierle, Bénédicte Charrier, Nino F. Läubli

**Affiliations:** 1Morphogenesis of Macroalgae, Laboratory of Integrative Biology of Marine Models, Station Biologique de Roscoff, CNRS, Sorbonne University, 29680 Roscoff, France; 2Ecology of Marine Plankton, Laboratory of Adaptation and Diversity in the Marine Environment, Station Biologique de Roscoff, CNRS, Sorbonne University, 29680 Roscoff, France; 3Molecular Neuroscience Group, Department of Chemical Engineering and Biotechnology, University of Cambridge, Cambridge CB3 0AS, UK

**Keywords:** microfluidics, microfluidic cultivation, brown alga, *Saccharina*, embryogenesis, lab-on-a-chip, microsphere tracking, kelp, laminariales, blue light, immunocytochemistry, time-lapse microscopy

## Abstract

The culturing and investigation of individual marine specimens in lab environments is crucial to further our understanding of this highly complex ecosystem. However, the obtained results and their relevance are often limited by a lack of suitable experimental setups enabling controlled specimen growth in a natural environment while allowing for precise monitoring and in-depth observations. In this work, we explore the viability of a microfluidic device for the investigation of the growth of the alga *Saccharina latissima* to enable high-resolution imaging by confining the samples, which usually grow in 3D, to a single 2D plane. We evaluate the specimen’s health based on various factors such as its growth rate, cell shape, and major developmental steps with regard to the device’s operating parameters and flow conditions before demonstrating its compatibility with state-of-the-art microscopy imaging technologies such as the skeletonisation of the specimen through calcofluor white-based vital staining of its cell contours as well as the immunolocalisation of the specimen’s cell wall. Furthermore, by making use of the on-chip characterisation capabilities, we investigate the influence of altered environmental illuminations on the embryonic development using blue and red light. Finally, live tracking of fluorescent microspheres deposited on the surface of the embryo permits the quantitative characterisation of growth at various locations of the organism.

## 1. Introduction

While the level of knowledge about the biology of marine organisms lags far behind that of terrestrial organisms, climate change accentuates the need to shift gears to fill these gaps. Indeed, a large number of marine organisms are under increasing threat [1,2] and further insights into how these organisms reproduce, develop, and grow are necessary to develop appropriate protection strategies [3]. Algae are one of these organisms, as they are the oceans’ primary producer and are especially sensitive to increasing temperatures [4]. Multicellular algae thrive along all coastlines of our planet from the tropics to the polar circles, where they are at the top of the local trophic chain [5]. The kelp, a group of brown algae, grow to lengths of several metres within a few months, which makes them the most conspicuous seaweeds. Furthermore, several kelp species, such as *Saccharina species pluralis*, are widely cultivated in Asia and western Europe for their economic value as a source of food and of hydrocolloids useful in industry [6]. Yet, how they develop is only scarcely known and, especially, the mechanisms that regulate the early steps of their embryogenesis, which are essential for both our understanding of the developmental processes and initial advances towards aquacultures, are a scientific black box [7]. One of the main reasons for this lack of insight is the difficulty of imaging developing organisms in a dynamic way that permits us to obtain accurate and quantitative information at the tissue and cellular levels. While the early embryo is a monolayered cell sheet [8], its upward growth prevents the acquisition of focused images of the entire embryo structure. Therefore, we introduce an experimental protocol enabling the cultivation of the embryos of the kelp *Saccharina latissima*, also known as sugar kelp, in horizontal confinement. Eggs and zygotes were loaded into the microfluidic chip forcing further growth into only two spatial dimensions. Coupled to time-lapse brightfield and fluorescence microscopy, this approach permitted us to quantitatively monitor each step of the development of the embryos in a dynamic way and in different culture conditions, including the influence of liquid renewal for on-chip cultures. We evaluated the usability of microfluidic growth chambers through the study of the specimens’ developmental stages and compared their performance to samples cultured in standard open-space in vitro environments using lab Petri dishes. Furthermore, by building on the improved optical monitoring capabilities, we quantify the effect of varying environmental illumination by analysing changes in the embryonic development induced through the application of blue and red light. Finally, we demonstrate the amenability of microfluidic devices to monitor local variations in growth through confocal microscopy of on-chip calcofluor staining, the tracking of fluorescent microspheres on the surface of the embryos, as well as the immunolocalisation of the specimen’s cell wall. Altogether, these tools demonstrate the suitability of microfluidic environments for the detailed investigation of embryonic growth, cell growth, and cell division patterns within the monolayered cell sheet of the embryo of the kelp *Saccharina latissima*.

## 2. Results and Discussion

### 2.1. Characteristics of S. latissima Development and Growth

*Saccharina latissima* is a prominent marine model organism of interest for developmental studies due to its heteromorphic diplophasic life cycle, as shown in Figure 1a [9]. Its diploid sporophyte grows following the successful fertilisation of the egg via a sperm cell. Several parallel cell divisions of the zygote produce an embryo that first consists of a stack of cells, before widening by perpendicular cell divisions to form a two-dimensional monolayered cell sheet called a lamina or blade. This morphology persists for several days until the embryo reaches a size of about one thousand cells, after which the cells start differentiating and initiate the thickening of the lamina by tilting their cell division plane into the third spatial axis.

This noteworthy sequential and tiered growth, further enhanced by the morphological simplicity of *S. latissima*, makes it a suitable model to investigate the development of brown algae. Additionally, the cell arrangement of the embryo is particularly original, as it depicts a grid in which cells are in contact with their neighbours by means of “four-cell junctions” [10]. This grid or “chocolate bar”-type of cell organisation is not observed in plants and animals under normal conditions as intrinsic mechanisms prevent their formation [11,12].

Given the repeated change in its dimensional growth, the detailed study of *S. latissima* embryos relies on the availability of methods suitable to assess the whole embryonic development in a single plane. However, *S. latissima* embryos, when cultured in open environments such as the sea or in vitro Petri dishes, grow mainly upright—including twisted and curved specimens—with only 9.7% of embryos growing parallel or nearly parallel to the bottom of the Petri dish. This makes optical microscopy-based observations of the whole embryo, especially in combination with time-lapse imaging, almost impossible and prevents accurate assessments of their growth rates or the mapping of cellular features within living embryos. Figure 1b shows an embryo growing in 3D while being attached to the glass bottom of a Petri dish with neither its stipe nor its lamina being fully in focus, demonstrating the limited accessibility for optical evaluations. Hence, the development of novel protocols permitting the detailed observation of the whole embryo by restricting growth to a single focal plane is crucial to further our understanding of the organism’s development and growth behaviour.

In addition, the in vitro cultivation of *S. latissima* embryos is further complicated due to the release of cues by the algae which reduce the reproduction success, as reported in [13]. Given the reduction in the liquid volume within microfluidic growth chambers compared to open-space cultivation, we, therefore, investigated how the density of *S. latissima* cultures impacts the production of eggs. We inoculated Petri dishes with five different concentrations of a mixture of short male and female gametophyte filaments (10, 25, 50, 75, and 100 µL in 2 mL of sea water, which relate to concentrations ranging from 1.26 to 12.6 gametophytes per µL). Figure 1c and Appendix A show that, while the number of eggs per gametophyte did not significantly change (red), increasing the density of gametophytes (to more than 1.26 gametophytes per µL) significantly reduced the rate of egg releases per total number of female gametophytes cells. Additionally, while no egg releases were observed at higher concentrations, ongoing maturations of gametophytic cells into oogonia were still monitored for all concentrations. In fact, oogonium differentiation was reverted into vegetative cells, as the cells started to divide and produce additional vegetative filaments (Appendix A). Altogether, this suggests that *Saccharina latissima* gametophytes negatively control egg differentiation and release in high culture densities, which is potentially caused through the secretion of growth-inhibitory molecules in agreement with [13]. Hence, this result highlights that the number of gametophytes doomed to be inoculated inside the chips must be small enough to ensure the production of a sufficient number of embryos available for observation.

### 2.2. Microfluidic Cultivation of S. latissima Embryos

Due to its ease of fabrication and its biocompatible properties, polydimethylsiloxane (PDMS)-based microfluidics is of increasing relevance for developmental biology [14] and has been widely applied for the investigation of various small organisms ranging from plant specimens [15,16,17] to animal models [18,19,20,21,22]. Furthermore, microfluidic-based cultivation has recently been reported as a suitable tool for the study of the filamentous growth of the brown alga *Ectocarpus silicosus* [23].

Here, we propose a PDMS-based microfluidic device capable of limiting the specimen’s growth to a single focal plane while minimising the potential limitations induced by nearby channel walls or similar constrictions (Figure 2a,b). As detailed in the Materials and Methods section, the device consists of a single-layer microfluidic chamber with a height of 25 µm to accommodate *Saccharina* embryos with an average thickness of approximately 15 µm. Given the chamber’s large in-plane dimensions, a grid of micropillars is used to prevent the chamber from collapsing, while also simplifying the loading procedure through the increased availability of surface area to which the specimens can attach. The PDMS structures are fixed to glass bottom Petri dishes before being filled and submerged in sea water. The application of Petri dishes reduces the chance of contaminations through the additional cover of the sample via a lid and allows for the simplified transfer of the specimens between incubation chambers and microscopy setups, such as those used for extended time-lapse imaging under controlled environmental conditions (Appendix A).

To assess the suitability of microfluidic cultivation, the chips were inoculated with different types of algal material, i.e., (i) unfertile and (ii) fertile female and male gametophytes, as well as (iii) female gametophytes with zygotes. When inoculated with gametophytes previously induced under white light for one week, the differentiation of eggs and sperm cells took place as early as 1 day after inoculation (see egg release in Appendix A), which is similar to what has been observed when the material was cultured in open environments [24]. In contrast, when fertile gametophytes were inoculated, the first zygotes were observed 1 to 2 days after inoculation. Finally, inoculated zygotes initiated embryogenesis immediately after inoculation. Imaging the embryo’s contour and the cell arrangement was easy as the lamina was located horizontally (Figure 2c). Time-lapse microscopy further showed that the cell division pattern within the embryo lamina and the formation of rhizoids from the base of the embryo took place as observed in open environments (Figure 2d and Appendix A).

We further assessed the maximum duration of living embryos inside the chip. Although the embryos remained alive during the whole 12-day monitoring period, a decrease in pigmentation was observed after 9 days (data not shown). Hence, it is worth noting that the monitoring of *S. latissima* growth within these chips should not be extended beyond 9 days, which, however, is sufficient for the detailed evaluation of its embryonic development.

Additionally, to prevent a lack of nutrients caused by an increase in culture density while simultaneously diluting potential growth-inhibitory molecules released by the neighbouring specimens as described previously (see Figure 1c and Appendix A), the use of a continuous flow or periodic liquid renewal was investigated. As the application of a continuous liquid exchange via a hydrostatic pressure difference has been found practically unfeasible (see Materials and Methods), the use of a pump is recommended. While the presented work relied on a pressure controller, in contrast to a passive solution, pumps have the advantage of permitting periodic actuations, which enables the use of a broad range of affordable, non-task-specific equipment, such as peristaltic pumps, even for microfluidic applications. To allow for continuous liquid exchange while preventing potentially damaging hydrodynamic forces, the pressure controller was set up to renew the volume of the medium inside the chip roughly 100 times per day.

We demonstrated that the proposed microfluidic chips are capable of providing environmental conditions suitable for the growth of *Saccharina* embryos in vitro, including cell growth, differentiation, e.g., for rhizoid formations, and growth of the laminae. However, it is worth highlighting that the current flow control and system performance could be further enhanced by replacing the applied system with higher-end options such as dedicated microfluidic syringe pumps. In the following sections, we assess whether the conditions produced by the confined environment of the microfluidic device allow for the maintenance of the early developmental transitions from the egg to the formation of an early embryo and, hence, their study using state-of-the-art microscopy techniques.

### 2.3. Assessment of Saccharina Developmental Steps inside Microfluidic Devices

The growth of *Saccharina* starts with several identified embryogenetic steps [8]. These steps are essential for the specimen’s development as they ensure the establishment of the two main body planes of the adult algae, i.e., the apico–basal and the medio–lateral axes. However, thus far, the effect of increasing temperatures and lower pH conditions induced by climate change or the impact of chemicals and drugs on the specimen’s development are unknown. Hence, the interest of these chips for further studies aiming to explore the possible consequences for the embryogenesis of *Saccharina latissima* depends on the chip’s amenability to allow *Saccharina* to undertake and achieve these embryogenetic steps. Therefore, we performed a comparison of the main developmental steps observed between the algal specimens growing inside microfluidic devices and outside the PDMS structure, i.e., in the same glass bottom Petri dishes. The embryogenetic steps of *S. latissima*, as introduced in Figure 1a, are 1. the egg release, 2. the fertilisation of the egg by the sperm cells, 3. the first, parallel cell divisions of the zygote leading to a stack of ~8 cells, and 4. the widening of the embryo by changing the planes of the cell division to 90° from the initial orientation.

Figure 3 shows that all of the abovementioned steps were successfully conserved within the device. The release of eggs inside microfluidic chips has been observed repeatedly (Figure 3a) and the eggs were successfully fertilised (Figure 3b). However, the very first cell division of the zygote was rarely observed for on-chip cultures as several eggs struggled to pass this developmental stage. This can be explained by the weak attachment of the eggs to the female gametophytes, which only consists of their flagella [25], in combination with substantial shear forces and high peak flow rates (estimated at 10^1^–10^2^ mm s^−1^) inside the chip during the inoculation. To circumvent this issue, vegetative gametophytes can be loaded to have eggs differentiated and released into the device long after the inoculation. Another observation is that zygotes managing to induce the first cell division often fail to elongate and, hence, to break the initial radial symmetry of the eggs.

For the abovementioned reasons and if the devices are only intended to allow for the detailed investigation and imaging of laminae, it is recommended to inoculate either elongated zygotes or embryos, which were shown to thrive within the device and consistently developed into large 2D monolayer laminae (Figure 3c and Appendix A).

In addition to the evaluation of the different developmental stages, the impact of the microfluidic device on the specimen’s growth behaviour has been investigated. Comparing the growth rate inside the device with and without liquid renewal as well as outside the device, i.e., in the same Petri dish next to the PDMS structure, provided us with clues about the general health of the specimens growing inside the device and allowed for the detection of possible drawbacks, e.g., in terms of access to nutrients. To quantify the effect of the device onto the growth of the embryo, we measured the apico–basal growth rate, i.e., the daily change in distance between the stipe and the apex of the lamina in both the specimens growing inside the chips and specimens growing in an open environment. For the latter condition, only specimens with a horizontal or near-horizontal orientation were considered to reduce measurement errors induced by the out-of-plane growth.

As shown in Figure 3d, the embryos inside the microfluidic devices with liquid renewal (blue, n = 15) kept growing qualitatively similar to the specimens outside the chip (green, n = 15), although, due to the low growth rates, no significant differences in the total blade growth between consecutive days were detected for specimens with liquid renewal following day 3 (or throughout the experiment for specimens without liquid renewal). Additionally, the growth dynamics of the specimens in devices with liquid renewal were similar to those of specimens growing in open space (Appendix A), with no significant differences in growth rates being observed within individual conditions for consecutive days. However, as indicated, the overall growth rate inside the device was, even with liquid renewal, substantially lower throughout the growth period and the differences in growth rate became significant after day 2, which further led to significant differences in the total blade growth from day 3 onwards. Interestingly, the length–width ratio (Appendix A) of the on-chip specimens with liquid renewal and samples growing in open space were not significantly different, while the outliers with higher ratios in the open environment can be accounted for by occasional twistings of the blades which reduce their width. The generally observed lower growth rates inside the chip could be caused by a reduction in light intensity due to absorption by the PDMS layer [26]; however, additional effects caused by the spatial confinement cannot be excluded. Furthermore, it is important to note that the on-chip growth rates and growth behaviours were only compared to specimens growing (near-)horizontally within the same Petri dishes, which suggests that variations between the observed conditions caused by gravitational effects can be neglected. However, gravimetric factors, such as gravitropism, which have been argued to influence growth hormones, e.g., auxin, in algal development need to be considered when comparing developmental data obtained via horizontally constricting devices, such as our microfluidic setup, to other literature [27,28]. Additionally, while our approach might be suitable for gravimetric investigations by mounting the chip vertically onto an appropriate microscopy setup, further design adjustment might be required, such as the re-orientation of the chamber axis to enable horizontal liquid renewal and prevent hydrostatic pressure differences or a reduction in chamber width to ensure equal distribution of nutrients and reliable medium exchange.

Finally, the applicability of the chip for physiological studies has been tested. Sea water filters longer wavelengths more than shorter ones, resulting in red light being unable to penetrate water columns deeper than 2 metres, while blue light is present up to 200 metres deep depending on the turbidity of the water. Brown algae live attached to the bottom of the sea at depths of 1 to 40 metres along coastal regions and several studies have already shown the developmental response of *Saccharina* to blue light at different stages of its life cycle and especially for the growth of the gametophytes and the induction of fertility [29,30]. Here, we tested the suitability of the chip to investigate the impact of blue and red lights on the development of the early embryos of *Saccharina*. *Saccharina* embryos were grown in the chip exposed to either red or blue light (Appendix A) for five days. Figure 4a,b,g,h highlights the difficulties concerning the evaluation and imaging of embryos growing outside the chip, as illustrated by the slices of strong red fluorescence (chloroplast autofluorescence) cutting through the embryos. Embryos within the chip displayed different morphologies depending on whether they were growing under blue (Figure 4c–f) or red (Figure 4i–l) light. Embryos under blue light displayed distorted morphologies characterised by a higher length/width ratio (Figure 4d–f) and several growth axes (Figure 4d). In contrast, embryos grown under red light (Figure 4i–l) displayed a morphology similar to the one under white light (see Figure 2c,d). These results not only strengthened the conclusion by Wang et al. based on experiments with embryos grown in open environments (glass slides in a flask or Petri dish), which reported higher length-to-width ratios under blue light [31], but also showed that our device provides images permitting accurate measurements by forcing the *Saccharina* blades to grow horizontally within the chip. The amenability of the chip design for confocal microscopy will also open the way for deeper studies concerning the role of the blue light receptor Aureochrome [32] during *Saccharina* embryogenesis.

### 2.4. Cell Biology Protocols to Accurately Image the Cells of the Kelp Embryos

Increasing our knowledge of the mechanisms involved in the embryogenesis of kelp requires not only quantitative values of the growth rate, such as those obtained by time-lapse brightfield microscopy, but also cell characterisations at the sub-cellular level. The use of transgenic lines expressing fluorescent proteins would be the ideal approach, as they permit us to monitor the localisation of proteins over time in living algae. However, while the expression of reporter genes was reported in *S. japonica* via a biolistic approach [33], it has never been repeated and, since, doubts about this approach in *Saccharina* were raised [34]. Therefore, we assessed the efficiency of several cell biology protocols that were previously developed for the filamentous brown alga *Ectocarpus* [35,36].

First, the cell wall was labelled with calcofluor white. Cellulose, a rigid polymer of β(1–4) glucose reported in brown algae’s cell walls at a level of approximately 10% [37], can be stained by calcofluor white, that is a vital fluorescent probe that binds to β-glucans [38]. We used it to stain *S. latissima* gametophytes and embryos inside the chip. Figure 5a,b displays a uniformly labelled embryo, as a representative of the whole population inside the chip. In addition, we did not observe any loss of specimen during this experimental procedure, supporting the fact that the gentle washing steps were compatible with this protocol. Interestingly, as calcofluor white is a vital stain, monitoring the fluorescent signal over time makes studies of the growth dynamics within the chip possible.

As the microfluidic device restrains growth to a 2D plane, it further allows us to obtain accurate outlines of the embryo shape and the precise contour of cells with calcofluor labelling. To add temporal and spatial precision to the growth mode of the algal specimen, we used fluorescent microspheres that stick to the cell surface. Microsphere-based tracking has previously been proven suitable to study growth in microfluidic environments, e.g., by Shaw et al. for the investigation of *Medicago truncatula* root hairs [39]. Additionally, the same beads have successfully been applied for the study of the brown alga *Ectocarpus* [36]. Hence, measuring the relative displacement of a microsphere over time in the same focal plane permits us to obtain quantitative data about the rate and direction of growth. The precision of the measurements directly depends on the size of the beads (down to 200 nm in the present study; see below) and on the parameters of the image acquisition (resolution down to ~200 nm in confocal microscopy). This high level of spatial accuracy should permit monitoring the trajectory of the beads, a necessary insight to further characterise the mode of cell growth [40].

Given the limited amount of detail currently known about the various developmental stages of the specimen and the cell wall of brown algae in general, different types of microspheres were tested, i.e., green-fluorescent carboxylate-coated spheres with a diameter of 500 nm and red-fluorescent amine-coated spheres with a diameter of 200 nm. Although the amine-coated red-fluorescent microspheres seemed to be adsorbed onto both gametophytes and sporophytic embryos, the carboxylate-coated green-fluorescent microspheres attached almost exclusively to gametophytes. This further demonstrates that *Saccharina*’s outer cell wall composition differs in gametophytes and sporophytes and that the type of beads has to be carefully chosen.

Using the derived parameters, we observed that the microspheres (pink dots) successfully attached to the outline of the embryos as shown in Appendix A. While the majority of particles were present around the blade, additional microspheres were present in other focal planes. Furthermore, both the shapes and the autofluorescence wavelength of the chloroplasts (Appendix A) show that the embryo is in good health. This latter observation further supports our conclusions based on the on-chip growth rates.

The resulting time-lapse monitoring of a growing embryo coated with red-fluorescent microspheres is shown in Figure 5c–e. The overlay of the two fluorescence microscopy images recorded at t = 0 h and t = 76 h display the rate of growth.

Finally, to assess the possibility to perform experiments of immunolocalisations within the chip, we used the BAM10 monoclonal antibody that recognises guluronates present in the cell wall of brown algae [37]. Figure 6 shows that, similar to embryos growing outside the chip (Figure 6a), BAM10 successfully labels the surface of early (Figure 6b) and late (Figure 6c) embryos within the device. In addition, in both conditions, BAM10 antibodies label the dome of tip-growing gametophytic cells (Figure 6a, right side) as well as rhizoids (Figure 6d, inside the device)—a pattern that is reminiscent of the BAM10 labelling of the apical cell of the filament of the brown alga *Ectocarpus* [41]. Hence, this work demonstrates that immunolocalisation experiments aiming at localising cell wall components in both embryo and gametophyte tissues of *Saccharina* are achievable inside the microfluidic chip. This opens avenues for the study of spatial distributions of molecular factors at the tissue and even organism scale, made possible thanks to the in-plane imaging conditions provided by the chip.

## 3. Conclusions

The in vitro investigation of marine specimens is crucial to further our understanding of their development and their interaction with these complex environments. Here, we report the first use of microfluidic chips for the study of kelp, specifically *Saccharina latissima*. While these algae are usually large-sized organisms growing upwards in the sea, we successfully demonstrated the suitability of microfluidic devices to monitor the initial steps of their embryogenesis within a chip restricting their amplitude of growth to 25 µm in height. This was made possible due to the specimen’s early-stage embryogenetic development, which leads to the formation of a monolayered cell sheet of only 15 µm thickness. We showed that, in this confined environment, *Saccharina* realises the main embryogenetic steps in conformity with the observations made in open environments. Furthermore, building on the investigation capabilities enabled by the microfluidic device, we qualitatively analysed the influence of varying illumination conditions, i.e., blue and red light, onto the specimen’s embryonic development. Finally, together with the in vivo labelling of the cell surface and on-chip immunolocalisation of cell wall components, this chip enables accurate observations of the specimen’s development over time, which provides new possibilities to study the embryogenesis of this organism in more detail and will help boost further knowledge increase for this important class of marine organisms.

## 4. Materials and Methods

### 4.1. Production of Saccharina Embryos: General Conditions (Genotype, Storage of the Gametophytes)

The *Saccharina latissima* embryos were produced from one isolated female gametophyte and one isolated male gametophyte, which were both derived from the same sporophyte, as described in [24]. In brief, female and male gametophytes were cultivated in PES1X (Provasoli-Enriched Seawater, see [42]) medium at 13 °C under red light, with a light intensity of about 16 µE s−1 m−2 (lower than 30 µE s−1 m−2 in all cases) with a long photoperiod of 14:10 (L:D). This maintained the gametophytes in a vegetative state. Gametogenesis was induced by transferring the gametophytes to white light with the same conditions of temperature and light intensity as indicated above. Five to ten days later, the gametes were released and fertilisation occurred. PES medium was made with natural sea water pumped at ~30 m under the sea level, filtered at 100–101 µm to remove debris, and autoclaved. Provasoli’s solution (see concentration below) was then added to this sea water to make different PES media. A concentration of PES1X (300 µL of PES in 1 L of natural sea water) was used for the culture of the gametophytes, while PES2X (600 µL of PES in 1 L of natural sea water) was used for the culture of the sporophytes (embryos and adults).

### 4.2. Design and Fabrication of the Microfluidic Device

The microfluidic design consists of a single chamber with a length and width of 7 mm and 5 mm, respectively, with an inlet and an outlet with diameters of 1.2 mm. The inlet is connected to the chamber via a funnel to ensure the uniform distribution of sea water and nutrients and to prevent the trapping of air bubbles. Additionally, to increase chamber stability as well as to simplify the trapping of specimens during the loading procedure, the chamber is filled with a grid of pillars with a diameter of 50 µm and a spacing of 350 µm. To allow the devices to be submerged in sea water inside small glass bottom Petri dishes with diameters of 2 cm (Model No. 801001, Wuxi NEST Biotechnology Co. Ltd., Wuxi, China), the outside dimensions of the polydimethyl-siloxane (Sylgard 184, Dow Corning, Midland, MI, USA) structures are 7 mm by 11.5 mm. The mould used for PDMS replication was fabricated using single-layer photolithography. A silicon wafer was prebaked at 200 °C for 10 min before being cooled down to room temperature and used as a substrate. Photoresist (SU-8 25, MicroChem Corp., Newton, MA, USA) was applied with a maximum speed of 2000 rpm to achieve a thickness of 25 µm and soft baked at 65 °C and 95 °C for 3 min and 15 min, respectively, before being cooled down to room temperature on the hotplate. An exposure dose of 170 mJ/cm^2^ was applied to transfer the design pattern (Film Photomask, JD Photodata, Hitchin, UK) into the photoresist. The wafer was post exposure baked for 1 min at 65 °C and 3 min at 95 °C before being developed for 3 min 30 s (SU-8 Developer, MicroChem Corp.) and hard baked for 10 min at 150 °C. PDMS with a ratio of 1:10 was poured onto the mould and cured under vacuum at room temperature overnight before being cut into individual chips and punching the inlet and outlet with a 2 mm diameter sterile biopsy punch. The PDMS chips were then dusted with compressed air and Scotch tape (Model 7100069922, 3M, Saint Paul, MI, USA) before being exposed to oxygen plasma in a plasma cleaner (Model PDC-32G-2, Harrick Plasma, Ithaca, NK, USA) for 45 s together with the Petri dishes and being brought into contact to seal the chambers chemically. These chips in dishes were then filled and submerged with PES2X and sterilised overnight under UV lights before hosting embryos.

### 4.3. Chip Inoculation

The chips were inoculated with different types of algal material, which differed in their position in the life cycle. It was (i) a mix of unfertile female and male gametophytes, (ii) a mix of fertile female and male gametophytes, or (iii) a mix of gametophytes with early-stage embryos. However, for most of the lab-on-chip experiments presented in this article, early-stage embryos growing on female gametophytes were used. For all cases, after quickly and gently replacing the culture medium with 1 mL of PES2X, the bottom of a Petri dish containing embryos with fewer than 10 cells was softly scraped with a cell scraper (Model No 83.3950, Sarstedt AG & Co. KG, Nümbrecht, Germany). The embryos were then resuspended in the medium by gently shaking the Petri dish, and then carefully sucked with a 1000 µL pipette with disposable tips large enough to allow for a continuous flow of embryos. The specimens were directly injected in the inlet of the chip (punctured with a biopsy punch of 2 mm diameter) and introduced into the chip by a gentle flow. Slowly emptying the tip into the chip took up to 30 s and a gentle up and down movement—using the flexibility of the PDMS—helped the embryos flow further into the device. This operation was easier when assisted with a stereomicroscope. The inoculated chip was then submerged with PES2X and transferred back to a 13 °C culture chamber with a long day schedule (14:10) of white lighting of around 16 µE s^−1^ m^−2^, along with other embryos.

### 4.4. Hydrostatic Pressure-Based Liquid Renewal

The volume flow resulting from a constant height difference can be simplified as follows:Q = ∆P/Rh(1)
where Rh is the hydraulic resistance and the pressure difference ∆P can be expressed as ∆P = −ρg∆h with ρ, g, and ∆h being the volumetric mass, the gravitational constant, and the height difference, respectively. Additionally, the hydraulic resistance induced by the microfluidic device and the tubing with an inner diameter of 500 µm and a length of 30 cm can be approximated with 1012 kg m^−4^ s^−1^ and 2.5 × 1011 kg m^−4^ s^−1^, respectively. Hence, the resulting height difference needed for a sufficiently slow liquid exchange, e.g., suitable to renew the internal chip volume about once a day, is located in the µm scale.

### 4.5. Pump-Based Liquid Renewal

A modular pressure controller (Flow EZ, Fluigent, Le Kremlin-Bicêtre, France) was used to allow for continuous medium renewal inside the microfluidic devices. Given the extended duration of the culture and to reduce the risk of contaminations, the pump was connected to the chip via platinum-cured silicone tubing with an internal diameter of 0.5 mm (Model No SHE-TUB-SIL-0.5*0.8, Darwin Microfluidics, Paris, France). The liquid renewal rate was estimated based on the excess media in the dish. 

### 4.6. Setting Up the Protocol for Time-Lapse Monitoring

The acquisition of images and time-lapse monitoring was performed using an inverted epifluorescence microscope (microscope: Leica DMI6000B, objective: Leica Microsystems HC PL APO 40x/0.85 CORR, camera: Leica DFC450C, software: LAS X 3.7) and an inverted confocal microscope (TCS SP5 AOBS, Leica Microsystems, camera: Leica DFC450C). The observations were made in the same lighting and temperature conditions as in the regular culture room. For this purpose, the microscope was placed in a room cooled to 17 °C and framed by a carbonate-glass chamber fitted to a thermostatically controlled air flow system (Model: CUBE & BOX, Life Imaging Services GmbH) to further cool the sample down to 13 °C. Two LED lamps were placed inside the chamber to provide adapted lighting (see Section 4.3 for detailed conditions).

### 4.7. Growth Rate Measurements

The growth was calculated by measuring the length and the width of the embryo, respectively the distance between the apex and the base of the embryo, and largest width, every 24 h in embryos growing (i) within the chips with flow, (ii) within the chip without flow, and (iii) in the Petri dish outside the chip. All conditions were applied in a single Petri dish, and 15 marked specimens for each condition were monitored for 7 days by time-lapse microscopy as described in 4.6. For long-term measurements outside the microfluidic device, only embryos growing horizontally were selected. Measurements were performed using the Fiji distribution of ImageJ2 [43], from which the total length (L), the length/width ratio, and the growth rate (L_n+1_ − L_n_)/(t_n+1_ − t_n_) were plotted as a function of time (t).

### 4.8. Duration of Embryos Viability in Microfluidic Devices

We assessed how long the embryos could stay alive in the microfluidic device by using four inoculated microfluidic devices loaded with 5 to 10 specimens each. These devices were placed in a 13 °C culture chamber with long day illumination (14:10). After opening the Petri dishes in a laminar flow hood, the devices were flushed manually every day with approx. 101–102 µL of PES2X using a 1 mL pipette. Embryo viability was evaluated for up to 12 days and viability was monitored through growth (or stagnation) as well as the colour of the embryos’ pigments.

### 4.9. Cultures and Assessment of the Impact of Culture Density

To assess the effect of the density of female gametophytes in the formation of eggs, a stock of female and male gametophytes was crushed into ~5- to 10-cell gametophyte filaments, mounted and optically inspected to ensure the required filament length, and 10, 25, 50, 75, and 100 µL of this mixture diluted in 2 mL PES1X (roughly 0.6 µL of PES per dish) were transferred to Petri dishes. After 2 weeks, the emerging eggs were imaged using an inverted microscope (Leica DMI-8). Each condition was observed in three Petri dishes and from two independent stock cultures. Counting was conducted from two snapshots taken on two different positions in the dishes. The number of eggs released was counted relative to the sum of dead + alive gametophytes as organisms and as their total cell number. The concentration of gametophytes per µL was derived as 252 (124 female GP and 128 male GP) using Malassez counting.

### 4.10. Assessment of the Embryonic Developmental Stages

The algal material was loaded in the chips and observed as described in Section 4.3 and 4.6. No flow was applied to the system for the duration of the observation. Time-lapse monitoring for 5 days allowed the observation of different transitions of the life cycle, especially those involved in the formation and development of *Saccharina* embryos.

### 4.11. Impact of Blue and Red Lights in Embryogenesis

Three-week-old embryos of Saccharina were inoculated in the chip with PES2X and cultivated for 5 days under red (9.2 μE⋅m^−2^⋅s^−1^) or blue (5.1 μE⋅m^−2^⋅s^−1^) light with a 16:8 (L:D) photoperiod. The light spectra are shown in Appendix A. Brightfield and autofluorescent embryos were observed in confocal microscopy (Argon laser, SP5, Leica Microsystems, Wetzlar, Germany). The autofluorescence of the chlorophyll was observed through a PMT with a 675–745 nm passing band.

### 4.12. Staining the Embryos with Calcofluor

Cellulose was stained by filling the microfluidic device with 20 µM calcofluor white solution (fluorescent brightener 28, F-3543, Sigma-Aldrich, St. Louis, MI, USA) diluted in PES2X. The incubation lasted 45 min at 13 °C (RT) in the dark. The solution was injected into the inlet of the device by pipetting. Observation was performed under UV light (excitation wavelength: 380 nm, emission wavelength: 475 nm) using epifluorescence (DMI8, Leica Microsystems) and confocal microscopy (405 nm emission laser, PMT band of 450–500 nm; SP5, Leica Microsystems) after flushing the channels with fresh PES2X at least twice and after incubation for at least 15 min at 13 °C between each rinsing step. No major bleaching effects, induced by the white light illumination needed for the cultivation of the specimen, were observed.

### 4.13. Marking the Embryo Surface with Fluorescent Beads

The embryos were marked with amine-coated 0.2 µm red-fluorescent microspheres (FluoSpheres™ ref. F8763, Invitrogen™, 580/605, Waltham, MA, USA). The concentration of the microspheres was found to be an important parameter for successful labelling, as the beads should uniformly label the specimens while still being distinguishable. Working solutions were prepared by diluting the beads to 0.05% in 500 µL of PES2X (*w/v*) in a 1.5 mL Eppendorf tube. Before dilution, the beads were scattered by 1 min of fast vortexing and 1 min of sonication (50 Hz, amplitude 40 Pa) with a probe cleaned with 70% ethanol. These working solutions were used to mark the surface of embryos grown in Petri dishes with diameters of 20–35 mm, as described below. Under a sterile laminar hood, the working solution was poured onto the embryo cultures previously emptied of sea water. In order to help the beads to attach to the embryos, the Petri dishes were carefully shaken in a circular motion for 1 min. The solution was then discarded and 1 mL of PES2X was added to wash off excessive beads during 1 min of circular motion. After five similar washing cycles, the dish was refilled with culture medium. These marked embryos were directly transferred into a chip for observation. Alternatively, they were placed back to the culture room.

Carboxylate-coated 0.5 µm green-fluorescent microspheres (FluoSpheres™ ref. F8813, Invitrogen™) were also tested, but they were only faintly attached to gametophytes and embryos.

### 4.14. On-Chip Immunolocalisation

The protocol was adapted from Charrier et al. [23]. The algal material in the chip was fixed for 2 ½ hours with 2% PFA prepared in H20. PFA was added as gently as possible to avoid too high pressures and distortions of the embryos inside the chip. Similarly, all mechanical pressures, e.g., applied manually directly onto chip, were avoided. All solutions were passed through the chip at least three times to ensure complete and homogeneous flooding of the chip volume. Primary and secondary antibodies were incubated for at least 2 h at 4 °C in the dark. The secondary antibody was coupled to Alexa488 and the signal was observed either in epifluorescence microscopy (DMI8, Leica Microsystems) with an FITC filter or with a SP5 confocal microscope (Leica Microsystems) with an Argon laser at 488 nm for excitation and through PMT with a band of 500–530 nm for emission. 

### 4.15. Statistical Evaluation

Statistical evaluations were performed in Prism 6 (GraphPad). The culture concentrations (Figure 1c) were analysed using a one-way ANOVA in combination with Tukey’s multiple comparisons test to detect differences between individual culturing conditions. Blade growth (Figure 3d), growth rate (Appendix A), and blade length–width ratios (Appendix A) were analysed using two-way ANOVAs, while Tukey’s multiple comparisons tests were applied to detect significant differences between the experimental conditions within days as well as between consecutive growth days within individual experimental conditions. Asterisks indicate the results (adjusted *p*-values) of the corresponding tests: * < 0.05, ** < 0.01, *** < 0.001, and **** < 0.0001. 

## Figures and Tables

**Figure 1 bioengineering-09-00718-f001:**
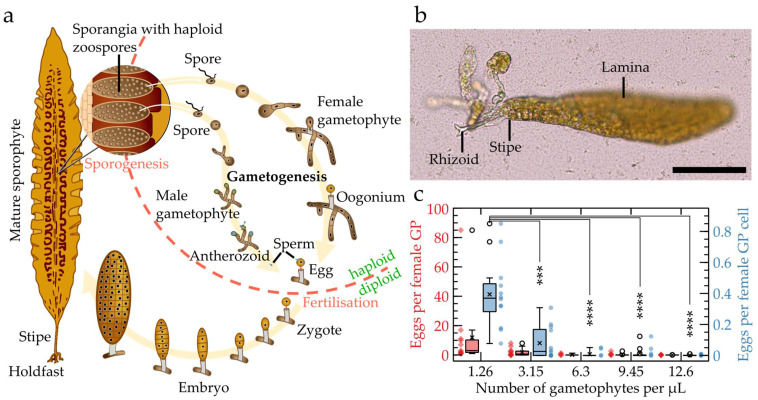
Biological specimen and density impact on culture. (**a**) Sketched life cycle of *S. latissima*. The mature sporophyte produces mobile zoospores that will develop into female and male filamentous gametophytes. Eggs hatched by the female gametophyte are then fertilised by swimming sperm cells released by the male gametophyte. Upon a series of parallel cell divisions, the resulting zygote develops into a multicellular embryo still attached to the female gametophyte before it develops its own attachment system first made of rhizoids (shown in 1b) and then a holdfast. (**b**) Optical microscopy image of a ~150-cell *S. latissima* embryo growing in a glass bottom Petri dish. Only the base, which is attached to the bottom of the Petri dish by the rhizoids, is in focus. The lamina is blurred as it is growing vertically within the water column. (**c**) Influence of culture density on egg release in open space expressed as the number of eggs per female gametophyte (GP, red) and number of eggs per total number of female gametophyte cells (blue). The number of eggs released by the female gametophytes is drastically reduced at higher densities of both female gametophytes and number of cells of female gametophytes. Asterisks indicate statistically significant differences relative to 10 µL of stock gametophytes analysed using one-way ANOVAs (see detailed protocol in Materials and Methods). No significant differences were detected for eggs per female gametophyte (red). Scale bar: 100 µm.

**Figure 2 bioengineering-09-00718-f002:**
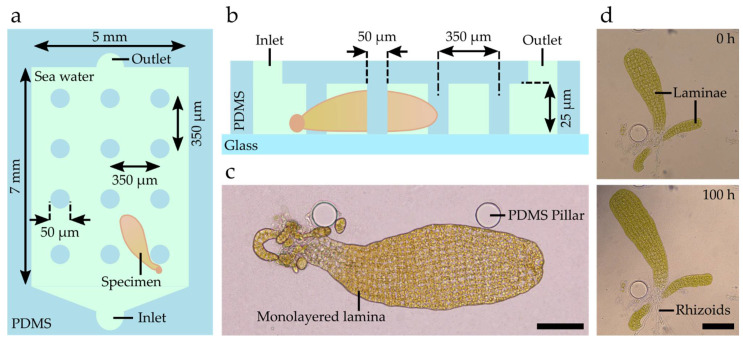
Microfluidic device enabling horizontally constrained growth of *Saccharina* culture. (**a**) Sketch of the microfluidic device (top view) showing the main chamber containing a grid of PDMS pillars as well as the inlet and outlet of the device. (**b**) Sketch of the side view of the microfluidic device with a constant chamber height of 25 µm and PDMS pillars with a diameter of 50 µm and a separation of 350 µm. (**c**) Brightfield image of a *Saccharina* monolayered lamina growing inside the device. This embryo was grown in a Petri dish before inoculation. All cells of the lamina are within a single focal plane. (**d**) Comparison of the same embryo growing inside the chip over a period of 100 h. Scale bars: 100 µm.

**Figure 3 bioengineering-09-00718-f003:**
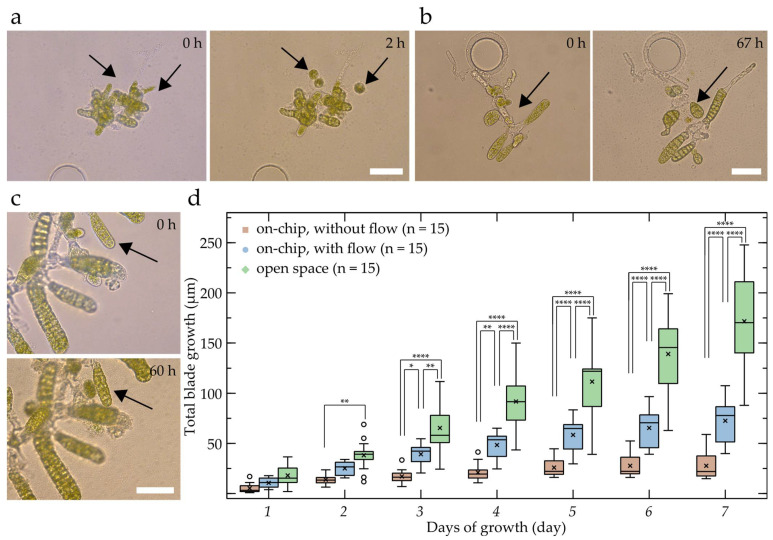
Assessment of the normal growth in the microfluidic device. Different developmental stages of *S. latissima* embryos observed while growing inside the chip: (**a**) induction of gametogenesis through egg release (three eggs have been released), (**b**) zygote formation after fertilisation and early embryo development (two cell divisions took place (arrow)), (**c**) first longitudinal cell division allowing growth and embryo expansion in the medio–lateral axis (arrow). (**d**) Growth rate comparison between embryos growing inside, with and without flow, as well as outside the chip. Statistically significant differences between the blade lengths of the different conditions within each day are indicated by the asterisks and were analysed using a two-way ANOVA (see Materials and Methods section). Scale bars: 50 µm.

**Figure 4 bioengineering-09-00718-f004:**
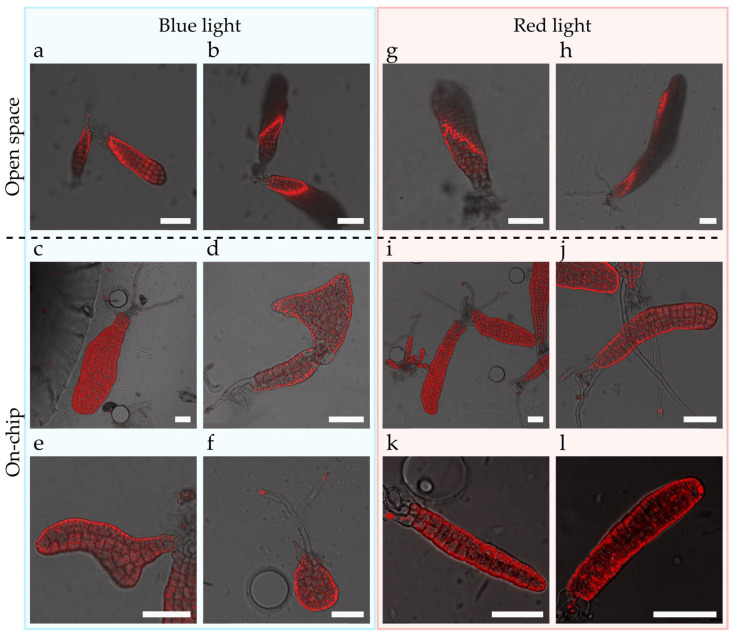
Morphogenetic impact of blue and red lights in the embryogenesis of *Saccharina*. (**a**,**b**,**g**,**h**) Confocal microscopy images of embryos grown outside the chips. (**c**–**f**, **i**–**l**) Confocal microscopy images of embryos growing inside the chips. (**a**–**f**) Embryos growing under blue light. (**g**–**l**) Embryos growing under red light. The embryos grown under blue light display distorted morphologies. Note that embryos with distorted shapes were also present outside the chip under blue light. Red: autofluorescence of the chloroplasts. Scale bars: 50 µm.

**Figure 5 bioengineering-09-00718-f005:**
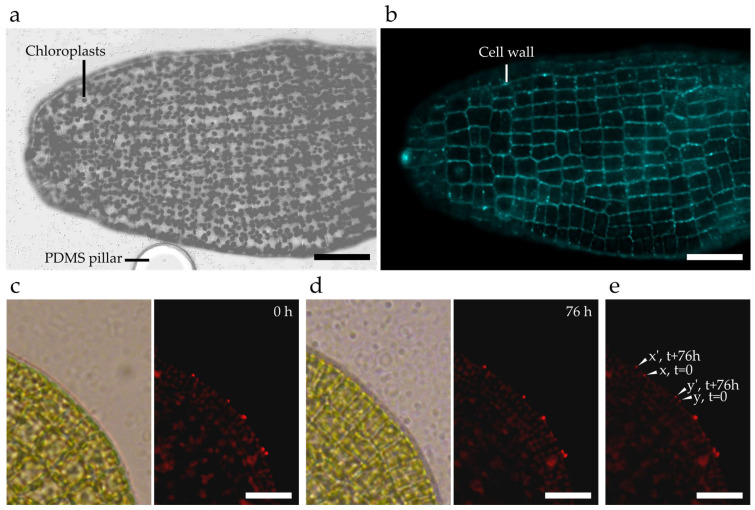
Microscopy-based imaging using on-chip cultures to track localised specimen growth. (**a**) Brightfield confocal microscopy image of an embryo blade with the auto-fluorescent chloroplasts (displayed here as grey dots). (**b**) Calcofluor white imaging of the same *Saccharina* blade. (**c**–**e**) Time-lapse tracking of fluorescent microspheres. (**c**) Brightfield and fluorescence microscopy images of functionalised red-fluorescent microspheres with a diameter of 200 nm were attached to the side of a monolayered cell sheet inside a microfluidic device on day 0. (**d**) The same location of the *S. latissima* lamina 76 h later. (**e**) Overlap of the two fluorescence microscopy images aligned. The time-lapse tracking highlights the difference in growth between the location of microsphere x (to x’) and microsphere y (to y’) for a growth period of 76 h. Scale bars: (**a**,**b**) 50 µm, (**c**–**e**) 25 µm.

**Figure 6 bioengineering-09-00718-f006:**
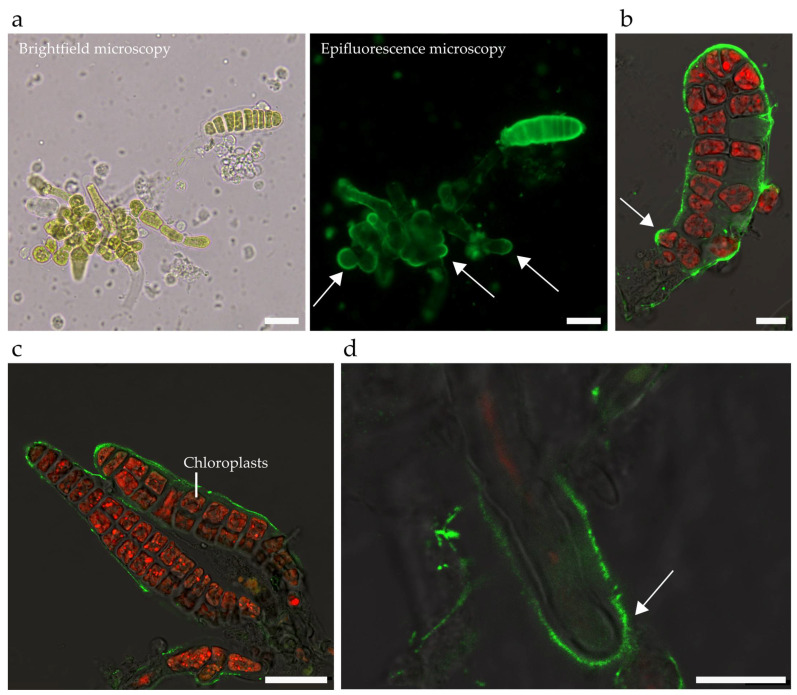
Immunolocalisation of guluronate alginates in the cell wall of *Saccharina* embryos. Different stages and tissues of *Saccharina* embryos are shown, from the zygote stage to the ~25-cell stage. (**a**) Labelled embryos grown in a Petri dish. An eight-cell-stage embryo showing a strong labelling in the outer cell wall is displayed (**left**: brightfield; **right**: epifluorescence), in addition to eggs, zygotes, and cells of the gametophytes showing labelling at the tip (white arrows). (**b**–**d**) Embryos growing within the microfluidic chip. (**b**) A ~15-cell-stage embryo labelled in the outer cell wall. Note the emerging rhizoid, which is strongly labelled (white arrow). (**c**) ~25-cell embryos showing a labelled outer cell wall. (**d**) Rhizoid showing an accumulation of guluronates in its dome (white arrow) and distal shanks. Green: Alexa-488 secondary antibody recognising the BAM10 primary antibody. Red: Autofluorescence of the chloroplast (displayed only in b, c, and d. Note that rhizoids have very few chloroplasts, which explains why the red autofluorescence signal is weak). The contrast has been enhanced linearly in b and c. Scale bars: (**a**,**c**) 25 µm, (**b**,**d**) 10 µm.

## Data Availability

The data presented in this study are available in the article and the Appendix A.

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
