# Peer review of "Cultivation and Imaging of S. latissima Embryo Monolayered Cell Sheets Inside Microfluidic Devices"

_bioengineering, 2022, doi:10.3390/bioengineering9110718_

Round 1

Reviewer 1 Report

This is an interesting and potentially useful device for experimentally controlled growth of thalloid algae using microscopic evidence by constraining the early growth to two-dimensions.

The authors provide adequate information on the design and application of the microfluidic device to allow informed decision making with respect to adoption and applications.

As the authors point out, constraining the growth laterally in two-dimensions in their device appears to decisively decrease the rate of growth, compared to growth unconstrained in three-dimensions, particularly after five days.

This particular bias of the instrumental effect  on growth does impose some possible limitations on making extrapolations to the natural environment about growth rates when experiments are done in the laboratory using this technique – however, the magnitude of the effect is sufficiently well documented in the manuscript.

Nonetheless, using a device such as this has a decidedly major advantage for documenting finer details of growth using the microfluidic chip as the authors demonstrate using varied staining and microscopic visualization methods.

With respect to the difference in growth in their preparation compared to three-dimensional vertical growth in a Petri dish, I wonder if the authors have considered the possible effects of constraints on gravimetric response (gravitropism) when growing horizontally, relative to the natural growth vertically. There is increasing evidence for the effects of growth hormones (e.g. auxin) on algal development and growth, including the effects of gravity on growth hormones in algae. Among these is one of the earliest studies by Davidson (1950) and more recently specifically examining the role of gravity (e.g., Sun et al., 2004) as listed below.

Davidson F. F. (1950) The effects of auxin on the growth of marine algae. American Journal of Botany 37: 502-510.

Sun H., Basu S., Brady S. R, Luciano R. L., and Muday G. K. (2004) Interactions between auxin transport and the actin cytoskeleton in developmental polarity of Fucus distichus embryos in response to light and gravity. Plant Physiology 135: 266-278.

It would be advantageous if the apparatus could be configured so the growth could be observed while the chip containing the growing alga could be oriented vertically and the angle, relative to the gravitational axis, varied; thus allowing applications where gravitational effects are taken into account. This of course would require additional adaptations of the microscope equipment to view the microfluidic device apparatus in a varied vertical position, etc., which also may imply some alterations in the construction and operation of the microfluidic device itself.

Overall, this appears to be a promising new laboratory experimental apparatus. The Supplementary information, including the videos, is useful.

I noticed only a small editing type of suggestion.

Line     Suggestion

363      “This allows the device to be submerged with sea water inside the small glass bottom Petri dishes ----”

Reviewer 2 Report

The authors present a microfluidic culture system to observe S. latissima embryo development by time-lapse microscopy over prolonged period of time thanks to the localization of the embryos in a quasi-2D system.

Major comments.

Given the microfluidic chip may be of interest in the field but is not particularly innovative, the paper value would be strongly increased by using the setup for a real case test scenario, like getting the temperature or pH range that preserve a certain phenotypic change during embryo development.

As the embryo development typically occurs upright, is there any gravity-missing effect on the horizontal system? Can you speculate on this if you do not have data?

I would skip a so detailed description of the hydrostatic pressure-induced flow, as the selected solution uses a peristaltic pump. 

Why using a peristaltic pump rather than a syringe pump that gives a better control of the flow rate and can get to very low flow rates?

As medium is changed hourly in the chamber, how can one be sure that the fluorescent bead position change is not due to flow-induced displacement of the specimen?

Any bleaching effect of fluorescent probes due to continuous system illumination? Any restriction on the usable fluorophore excitation wavelengths?

Minor comments.

- lines 212-219: long sentence, difficult to read

- Fig 3d caption: the difference is not present only on day 6 if I understand correctly (asterisk meaning is not defined)

- line 363: "The allow", something missing?

- line 430: why 1-mL pipette for such small volumes?

Reviewer 3 Report

Clerc et al. describes a very useful method to monitor and precisely measure the growth of microscopic sporophytes of S. latissima. However, the description of the methodology is not clear, and more information should be added in this section so that the readers easily understand how the experiments were performed. On the other hand, the Results & Discussion section has methodological details that should be moved to the Materials and Methods, requiring therefore rewriting.

In my opinion this manuscript can be suited for publication in Bioengineering after minor revisions.

Minor concerns, suggestions and questions are detailed below.

Results & Discussion

Line 97 - Figure 1a

I suggest changing the legend of figure 1a “The mature sporophyte produces female and male mobile zoospores that will develop into filamentous gametophytes.” to “The mature sporophyte produces mobile zoospores that will develop into female and male filamentous gametophytes.” It’s not possible to differentiate female and male zoospores, only after the germination into gametophyte sex differentiation is possible, therefore in kelp literature is not usual to write female and male zoospores, as it is not known which zoospore will develop into a male or female gametophyte. Similarly, in the figure a, I suggest removing female and male from spore designation.

Figure 1c

I recommend that the authors replace the Volume of stock gametophytes (µl) by gametophytes per ml or cm2, i.e., a known gametophyte density that will permit comparison between similar studies and allow the readers to use the correct gametophyte density for optimal egg release in their own studies.

Line 108

“Indicated significance is relative to 10 µl of stock gametophytes”. Which indicated significance? Are the asterisks showing significant differences relative to 10 µl of stock gametophytes? Please include this information in the legend. Why only statistical results are shown for the nº eggs per total nº of female gametophyte cells and not per female gametophyte?

 Line 111

“…in vitro cultivation of S. latissima embryos is further complicated due to the release of growth inhibiting cues by the algae which, as reported in [13], affects culture densities.” Please consider revising this sentence as Ebbing et al. (2020) suggested that suppressing reproduction cues such as hormones or autoinducers might be responsible by the negative effect of density on reproduction success.

Line 117

“…crushed embryos…” I assume you crushed gametophytes and not embryos, thus I suggest changing to “…crushed gametophytes…”.

Line 118

“…the rate of egg releases per cell of female gametophytes…” Please revise it, should be per female gametophyte or per total number of female gametophyte cells. Which one are you referring to?

Line 147

Replace “(see Figures 2a and 2b)” by “(Figures 2a and 2b)”.

Line 154

“Once the chambers are located with Saccharina material (see below), …” See below where? Please specify.

Line 158

Replace “(see Supplementary Figure S2)” by “(Supplementary Figure S2)”.

Line 160

“To assess the suitability of microfluidic cultivation, the chips were inoculated with different types of algal material, …” This experiment is not detailed in the Materials and Methods section. Please include.

Line 173

As far as I understood different methods for medium renewal were used for assessing the embryos viability in microfluid devices and for measuring the embryos growth rate in the same devices. For the embryos viability experiment, the medium was changed manually only once per day (line 429), while for the growth rate experiment the medium renewal was done through a pump system running 1.63 µl/min. So, how can the authors say that S. latissima growth can be monitored for 9 days based in an experiment performed under different conditions?

Line 179 to 193

This information should be moved to the Materials and Methods section.

Line 211

I was not able to find this experiment explained in the Materials and Methods section.

Figure 3b and c

The authors should also add the arrow corresponding to the initial structure in the picture of time period 0h, so the readers can easily identify the developmental change with time.

Figure 3d

A significant difference was only observed on day 6? Please include this statistical information in the graph. The * in the graph stands for what? Please include the explanation in the figure legend.

Line 264

“…growth rate inside the device was significantly lower at times, leading …” Please specify at what time periods the growth rate was lower.

Conclusions

Line 337

Replace “Furthermore. together with the …” by “Furthermore, together with the …” 

Materials and Methods

Line 343

“4.1 Production of Saccharina embryos: general conditions (genotype, storage of the gametophytes”, although genotype is mentioned in this subsection title no information is included. How many gametophytes/isolates were used? Derived from how many sporophytes?

Line 345

“…cultivated in PES1× medium…” Here was the first time PES was mentioned and not in line 350. The full name should be included in this sentence. The PES recipe used was the original or a modification, please include the reference.

Line 419

For how long the growth was followed? How frequently the medium of the Petri dishes was changed? How many embryos were measured per chip/Petri dish? And random selected embryos were measured through time or marked ones (always the same)? How many chips and Petri dishes were compared? The growth was evaluated from pictures, which camera was used? Indicate the formula used to calculate the growth rate.

Line 425

Please specify for how long the embryo viability was followed.

Which parameters were used to measure embryo viability?

Line 433

“… gametophytes was crushed into ~5-cell gametophyte filaments…” How was the size of the gametophyte filaments controlled? Did you pass through sieves to have gametophytes of similar size?

Line 435

How many ml of PES medium had each Petri dish?

Line 438

“The number of eggs released was counted relative to the sum of dead + alive gametophytes…” Why was the number of dead gametophytes taken into account? They will not release eggs…

Figure S3

“Asterisks indicate the result of statistical tests” The authors should explain that the asterisks indicate significant different between on-chip and open space (I assume). Please include this information in the legend.

Round 2

Reviewer 2 Report

The authors have addressed all my concerns.